# Influence of Different Addition Ratios of Fly Ash on Mechanical Properties of ADC10 Aluminum Matrix Composites

**Shueiwan Henry Juang * and Ching-Feng Li**

Department of Mechanical and Mechatronic Engineering, National Taiwan Ocean University, Keelung 202, Taiwan; poison0278@yahoo.com.tw
* Correspondence: shjuang@mail.ntou.edu.tw; Tel.: +886-2-24622192

**Abstract:** Aluminum-fly ash composites are formed by the chemical reaction between fly ash and the high-temperature aluminum-based alloy, which melts to form aluminum oxide as a reinforcing phase, which belongs to a composite of in situ synthetic reinforcing phases. Compared to aluminum-based alloys, composites have superior strength, rigidity, damping capacity, and wear resistance, but lower ductility and toughness. In this study, different fly ash addition ratios (0, 3, 6, 9, 12, and 15 *wt*%) were added to the ADC10-2Mg alloy melt via stir casting to form the aluminum-fly ash composite under the chemical reaction at 800 °C for 30 h. Subsequently, microstructure observation, density and porosity measurements, and hardness and tensile tests were conducted to analyze the influence of different fly ash weight percentages on the mechanical properties of aluminum-fly ash composites. According to the results, an aluminum-fly ash composite with good dispersibility of fly ash debris can be prepared by stir casting, and the fly ash particles gradually decomposed small debris as they reacted with the aluminum-based alloy at high temperatures during a long-term reaction process. The density of the aluminum–fly ash composite was reduced by adding fly ash, and its hardness and tensile strength were improved as well. However, the porosity increased with the amount of fly ash and the ductility was diminished. For the aluminum-fly ash composite with 6 *wt*% of fly ash, its density decreased by approximately 2%, the hardness and tensile strength increased by 7% and 49%, respectively, and the ductility decreased by 35%, as compared to those of the ADC10 alloy.

**Keywords:** fly ash; aluminum matrix composite; in situ synthetic reinforcing phase; stir casting





## 1. Introduction

Studies on metal matrix composites (MMCs) began in 1960 and have a history of over 50 years. Other materials have been added to the matrix to enhance the physical, chemical, mechanical, and electrical properties of the original material. Metal-based composite materials are widely used in automobiles, aerospace, and other industries because of their excellent strength, rigidity, damping, wear resistance, thermal expansion, and light weight. Early MMCs focused on improving performance; however, their costs were relatively high. In recent years, with the rise of environmental protection concerns of low cost, low carbon emissions, and low energy consumption, light metals, such as aluminum, magnesium, and titanium alloys, have been gradually used in industry to replace steel materials; however, the mechanical properties of light metals are far inferior to those of steel, which can be improved by adding reinforcing materials into the metal matrix to improve the mechanical properties of composite materials. High-performance products are lighter and exceed steel materials in certain applications.

Common reinforcing phases of metal matrix composites include silicon carbide (SiC), alumina oxide ($Al_2O_3$), and graphite (Gr). In general, SiC, $Al_2O_3$, and Gr are more expensive to use as reinforcing particles for aluminum matrix composites and these particles are especially suitable for industrial products with higher values [1,2]. However, aluminum matrix composites have several advantages, such as good strength, wear resistance, and

weight. Many studies have added industrial, agricultural, or other wastes to aluminum matrix materials to further reduce the cost of casting aluminum matrix composites. These developments have produced technology for the formation of aluminum-based composites, such as fly ash, bamboo ash, and straw ash. Some studies on aluminum-based composites have been published, among which bamboo and rice husk have significant potential and can be used as complementary reinforcement materials for the development of low-cost and high-performance aluminum hybrid composites [3–5]. Such studies have also created the value to recycle resources for waste reuse [6–9].

Fly ash is a type of reinforcement material with low density and low cost, and is available in large quantities as solid waste [10,11]. Several studies have shown that, compared with the base alloy, the composite material with fly ash particles, the so-called aluminum fly ash (ALFA) composites [12], significantly improve the mechanical properties, wear resistance, corrosion resistance, high strength-to-weight ratio, low value of thermal expansion coefficients, good thermal performance, higher damping capacity, and wear resistance when compared to matrix alloys [13–19]. Therefore, many industries have enhanced aluminum matrixes by adding reinforcement materials to meet their production requirements [16]. For example, aluminum-based composites are often used to manufacture pistons, cylinder liners, drum brakes, connecting rods, parts of vehicle braking systems, and Cardan shafts. Regarding the application of aluminum in automobiles, studies have shown that the composite technology allows the production of lightweight, low-cost, and high-performance brake discs. Moreover, owing to their good mechanical properties, low density, and high strength-to-density ratio, they are widely used in aviation, marine, and aerospace [20–24].

Fly ash is a coal-fired waste from thermal power plants and its main compounds are silica ($SiO_2$), alumina ($Al_2O_3$), and iron oxide ($Fe_2O_3$) [25]. The size of fly ash particles is between 0.5 and 400 µm categorized into three groups based on their morphology, that is, solid spheres, hollow spheres or cenospheres, and porous pellets [18], with a true density between 2.0 and 2.5 $g/cm^3$. The density decreases with the decrease in the iron oxide content [26,27].

Based on previous studies on aluminum-reinforced materials [28,29], the use of fly ash can optimize the mechanical properties of the base metal; however, these studies are applicable to some base metals such as aluminum alloys Al-8FA and AA6061, and aluminum 7075 alloys. In this study, the die-cast alloy ADC 10 was used as the matrix metal and 2 *wt*% Mg was added to enhance the wettability of the molten aluminum matrix and fly ash. During the preparation of the ALFA composite slurry and ingot, the treated fly ash particles were added to the ADC10 molten metal at 800 °C with mechanical stirring to prevent fly ash from agglomerating and suspending. The fly ash was reacted with aluminum in the matrix alloy for 30 h, and then the ALFA composite slurry was poured into the prepared mold for forming. Finally, the formed ingots were used to make test pieces to observe the dispersibility of fly ash in the matrix alloy ADC10, and microstructure analysis, density measurement, porosity calculation, hardness, and tensile tests were also performed. The main purpose of this study is to use the general-purpose die-casting alloy ADC10 as the matrix alloy to study the effect of the fly ash addition ratio on the physical and mechanical properties of ALFA composites.

The demand for the strength and toughness of automotive components is gradually increasing, such as the special die-casting alloy $AlSi_9Cu_3Mg$. As the chemical composition of this alloy is between those of ADC10 (good toughness) and ADC12 (good strength), the requirements of such alloys can be met as long as a small amount of Mg is added. If ADC10 can be used as the matrix alloy and an amount of Mg is added (contributing to the wettability of aluminum alloy and fly ash), the Si content of the matrix can be increased by reducing the Si by the reaction of fly ash and aluminum, and the reinforcing phase $Al_2O_3$ is generated simultaneously. The strength and toughness of the strengthened matrix alloy ADC10 can meet the requirements of automotive components without the use of expensive special die-casting alloys. Therefore, the specific contribution of this research is not only

to reduce industrial waste but also to create the value of waste reuse and improve the mechanical and physical properties of general products.

## 2. Materials and Methods

### 2.1. Method

The fly ash used in this study was produced by the "Yunlin Mailiao Power Plant" in Taiwan and belongs to the F-class fly ash, that is, $SiO_2 + Al_2O_3 + Fe_2O_3 > 70\%$. Pre-treatment of fly ash was sequentially performed as follows: (1) screening particle sizes between 53 and 106 µm; (2) using flotation to remove unburned carbon; (3) applying magnetic separation to remove the high iron content of the fly ash; (4) acid washing to remove the surface impurities; (5) high-temperature baking to remove the unburned carbon and impurities. As shown in Table 1, the ratio of $Fe_2O_3$ and loss on ignition (L.O.I) decreased significantly among the compounds of the fly ash after the treatment. The matrix alloy was selected from the die-cast aluminum alloy ADC10 in the JIS H5302. The chemical composition shown in Table 2 was measured by SGS Taiwan Ltd., and the average value was obtained from four measurements. Two *wt*% of pure magnesium was added to the matrix alloy to enhance the wettability between the aluminum matrix and fly ash particles because the magnesium content of the ADC10 alloy was only 0.3%.

**Table 1.** The compound phase of fly ash before and after treatment (*wt*%) data from [30].

| Condition | $SiO_2$ | $Al_2O_3$ | $Fe_2O_3$ | L.O.I. | Moisture | Others |
|---|---|---|---|---|---|---|
| Original fly ash | 58.9 | 25.5 | 4.93 | 4.10 | 0.2 | Bal. |
| Treated fly ash | 68.0 | 26.9 | 2.90 | 0.2 | 0.1 | Bal. |

**Table 2.** Chemical composition of die-cast aluminum alloy ADC10 (*wt*%).

| JIS | Si | Cu | Mg | Zn | Fe | Mn | Ni | Sn | Al | Cr | Ti | Ca | Pb | Zr |
|---|---|---|---|---|---|---|---|---|---|---|---|---|---|---|
| ADC10 | 8.755 | 2.290 | 0.230 | 0.778 | 1.448 | 0.177 | 0.037 | 0.032 | 86.130 | 0.061 | 0.028 | 0.001 | 0.029 | 0.004 |

In this study, the stir casting method was used to prepare aluminum-based fly ash composites with different fly ash ratios (0, 3, 6, 9, 12, and 15 *wt*%). In the preparation of the ALFA composites, molten aluminum matrix and fly ash were heated to 800 °C. The fly ash was added to the prepared crucible at a rate of 0.06 g/s under a stirring speed of 300 rpm to be stirred and mixed. The stirring configuration used in this study was an electric mixer, which is simple to operate and requires no additional instruments. As a predetermined ratio of fly ash was added, the composite slurry was poured into a square mold preheated to 150 °C. After it was completely solidified and cooled to room temperature, the ALFA composite block was removed from the mold and the prepared composite block was cut into small pieces of blocks, each weighing 500 g, for the following tests. The ratio of the MMC preparation is shown in Table 3, whereas the mixing system and equipment used in the experiment are shown in Figure 1.

**Table 3.** The preparation ratio of aluminum-based fly ash composite block (*wt*%).

| Ingot Number | ADC10 | Mg | Fly Ash |
|---|---|---|---|
| 1 | 98 | 2 | 0 |
| 2 | 95 | 2 | 3 |
| 3 | 92 | 2 | 6 |
| 4 | 89 | 2 | 9 |
| 5 | 86 | 2 | 12 |
| 6 | 83 | 2 | 15 |

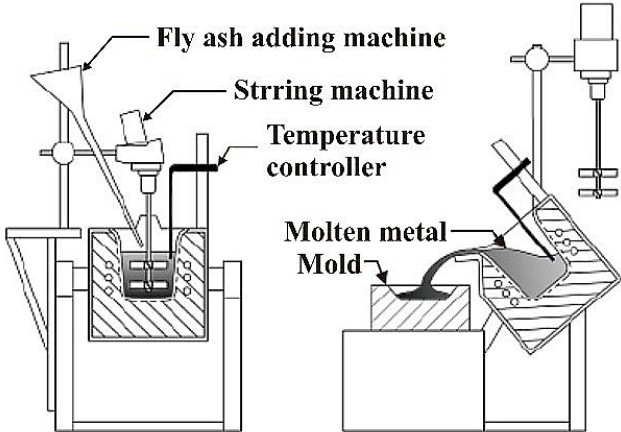

**Figure 1.** Schematic of the mixing system and equipment.

After the ALFA composite blocks were prepared, 500 g of the composite obtained from the prepared composite block was loaded into a graphite pot and placed in a high-temperature furnace, which is an induction heating apparatus type, to be heated to 800 °C for 30 h. As the density of fly ash is lower than that of the aluminum matrix, fly ash particles may float in the upper layer of the ALFA slurry during a long-term chemical reaction, resulting in an uneven distribution of fly ash particles in the aluminum matrix.

To overcome this phenomenon, the preparation steps of the experimental test specimen are shown in Figure 2: (1) first, reduce the temperature of the high-temperature furnace from 800 °C to 640 °C and continue to stir the slurry in the furnace until it reaches a uniform temperature for a period to avoid the floating of fly ash due to the long formation period of slurry; (2) remove the graphite pot from the high-temperature furnace, scrape the oxide layer on the top of the slurry to avoid it remaining in the casting to form defects, pour the slurry, and scrape off the oxide layer into the steel cup preheated to 180 °C; this takes approximately 30 s; (3) to evenly disperse the fly ash in the slurry, mechanically stir the mixture for 35 s at 150 rpm; (4) after completely stirring, immediately pour the slurry into the stainless-steel square mold (400 × 320 × 130 mm) preheated at 150 °C to form the test piece. After it is cooled, the production of the experimental test piece is completed. The ALFA composites prepared by this procedure ensure a uniform distribution of fly ash in the aluminum alloy melt [31,32].

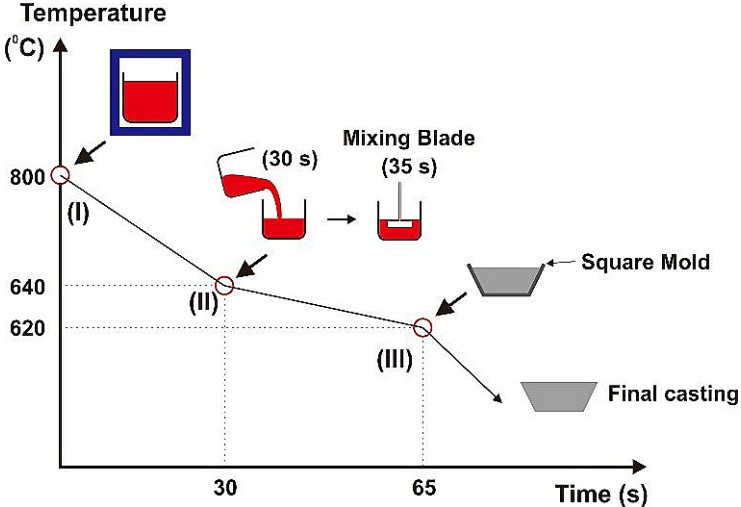

**Figure 2.** Schematic of the preparation steps of aluminum-based fly ash composites.

### 2.2. Material Testing

In this study, 500 g of each block was prepared, and the blocks were placed into a graphite pot and heated at 800 °C for 30 h to perform the chemical reaction. Microstructure observations, density measurements, porosity ratios, hardness tests, and tensile tests of the prepared ALFA composite test pieces were conducted to explore the effects of different fly ash addition ratios on the physical and mechanical properties of the ALFA composites. A detailed description of each test is as follows.

### 2.2.1. Microstructure Observation

The microstructure of each specimen was measured using an Olympus BX61 optical microscope (Olympus Scientific Solutions Americas Inc., 48 Woerd Avenue Waltham, MA, 02453, USA.) at 50 and 500× magnification. Then, the metallographic specimen was prepared using the CNC machining center to mill the observation surface (that is, section direction), and an abrasive paper with #100, #180, #400, #600, #800, and #1000 was used in sequence to grind the surface; finally, alumina powder polishing cloths containing 0.5, 0.3, and 0.1 μm were applied for polishing. The metallographic test pieces were completed when the observation surface was bright and scratch-free. Finally, the dispersibility of the fly ash, the bond between the fly ash and the aluminum matrix, and the reaction conditions were observed using an optical microscope.

### 2.2.2. Density Measurement and Porosity Ratio

The electronic balancer Precisa XS 365M was used to measure the weight of the test piece in water and air, and the actual density ($\rho_{mr}$) of the test piece was calculated using Archimedes' principle. The theoretical density ($\rho_{th}$) was calculated according to the weight percentage of alloying elements of the specimen and its atomic weight. Equation (1) was then applied to calculate the porosity ($V_{porosity}$) of the ALFA composite.

$$V_{Porosity} = 1 - \frac{\rho_{mr}}{\rho_{th}} \tag{1}$$

### 2.2.3. Hardness Test

Test specimens were prepared by sequentially grinding the bottom surface with abrasive papers #100, #180, #400, #600, #800, and #1000. According to ASTME10, a Brinell hardness tester (HB-3000C hardness tester, Laizhou Huayin Testing Instrument Co., Ltd., 215 Gulou Street, Laizhou, Shamdong, China) was used to perform the hardness tests. After the test, the indented diameter of the test piece was measured, and the measured data were input into the Brinell hardness Equation (2) to calculate the Brinell hardness, where $P$, $D$, and $d$ are the applied load (10 kN), indenter diameter (10 mm), and indented diameter (mm), respectively.

$$HB = \frac{2P}{\pi D(D - \sqrt{(D^2 - d^2)}} \tag{2}$$

### 2.2.4. Tensile Test

A CNC machining center was used to machine the ALFA composite block into the tensile test specimen, as shown in Figure 3. The tensile test piece was then ground using # 400 abrasive paper. According to the ASTM–E8 specification, a universal tensile testing machine (HT-9102 tensile testing machine from Shanghai Hongxing Ultrasonic Electronic Instrument Co. Ltd., 2678, Qixin Road, Minhang Qu 201100, China) with a capacity of 50 kN was used to perform the tensile test at a strain rate of $2.5 \times 10^{-4}$ s$^{-1}$.

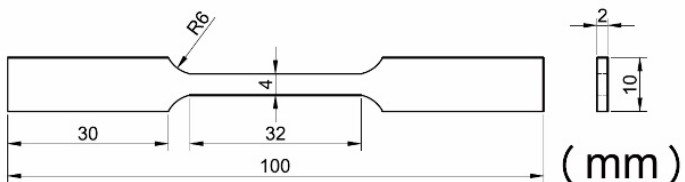

**Figure 3.** Schematic of the tensile test specimen.

## 3. Results and Discussion

### 3.1. Microstructure Observation

The interaction mechanism between fly ash and aluminum matrix was reported by Myriounis et al., in which they added magnesium to the matrix alloy mixed with fly ash and observed the formation of $Mg_2Si$ precipitates and quasi-binary $Al$-$Mg_2Si$, indicating that the fly ash particles were decomposed. Furthermore, in the authors' previous study, 3 *wt*% Mg was added to the mixture of 5 *wt*% of fly ash and liquid aluminum using the stir casting method, and after 40 h of reaction, the fly ash particles were observed to be broken into pieces and diffused into the liquid alloy, as shown in Figure 4 [33].

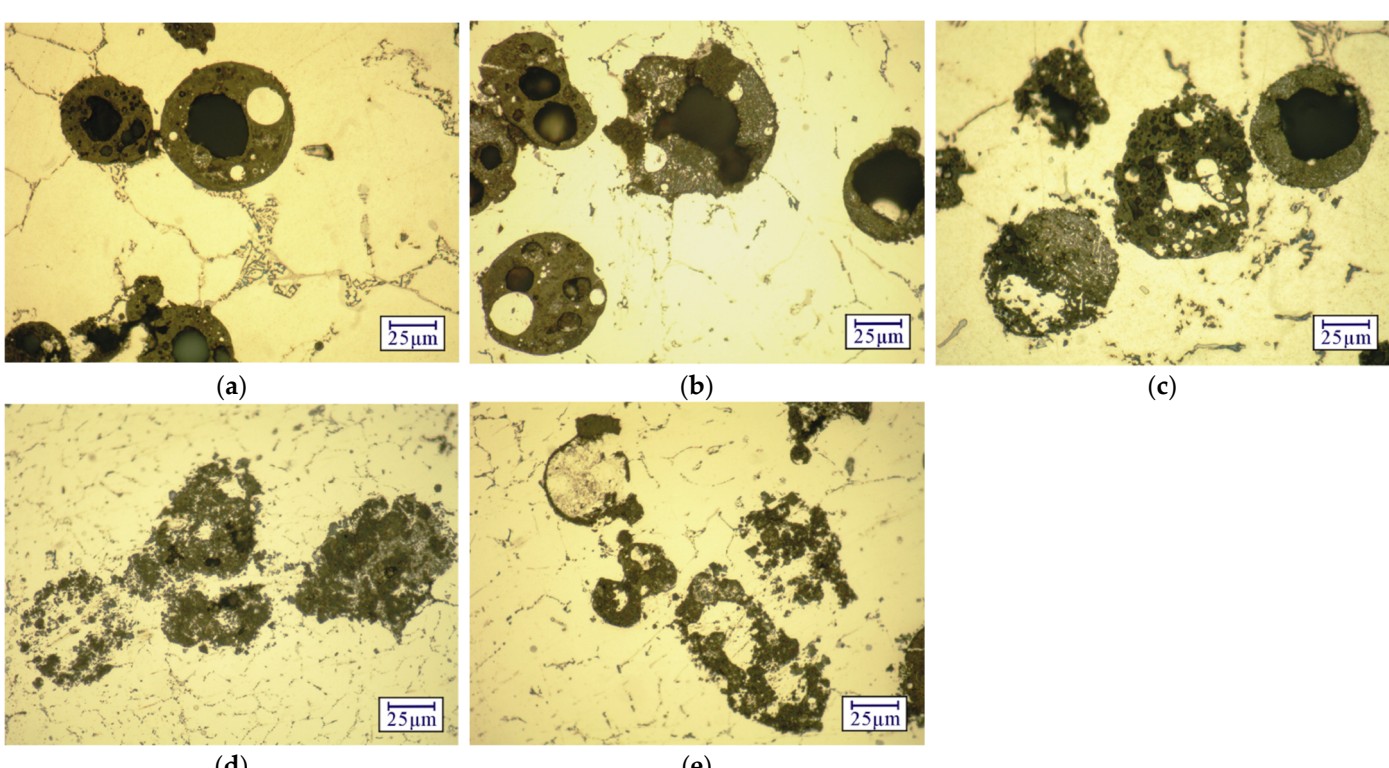

**Figure 4.** Optical micrographs of Al-3Mg/5 *wt*% fly ash composites after reaction at (**a**) 0 h, (**b**) 10 h, (**c**) 20 h, (**d**) 30 h, and (**e**) 40 h [33].

Optical microscopy was used to observe the ALFA composite specimens. Figure 5 shows an optical micrograph of the composite prepared with each fly ash addition ratio at 50× magnification. The black particles in the figure are fly ashes, which are shown for each addition ratio of fly ash. Fly ash particles had good dispersibility in the ALFA composite, meaning a composite with good dispersibility can be prepared by stir casting and subsequent chemical reactions.

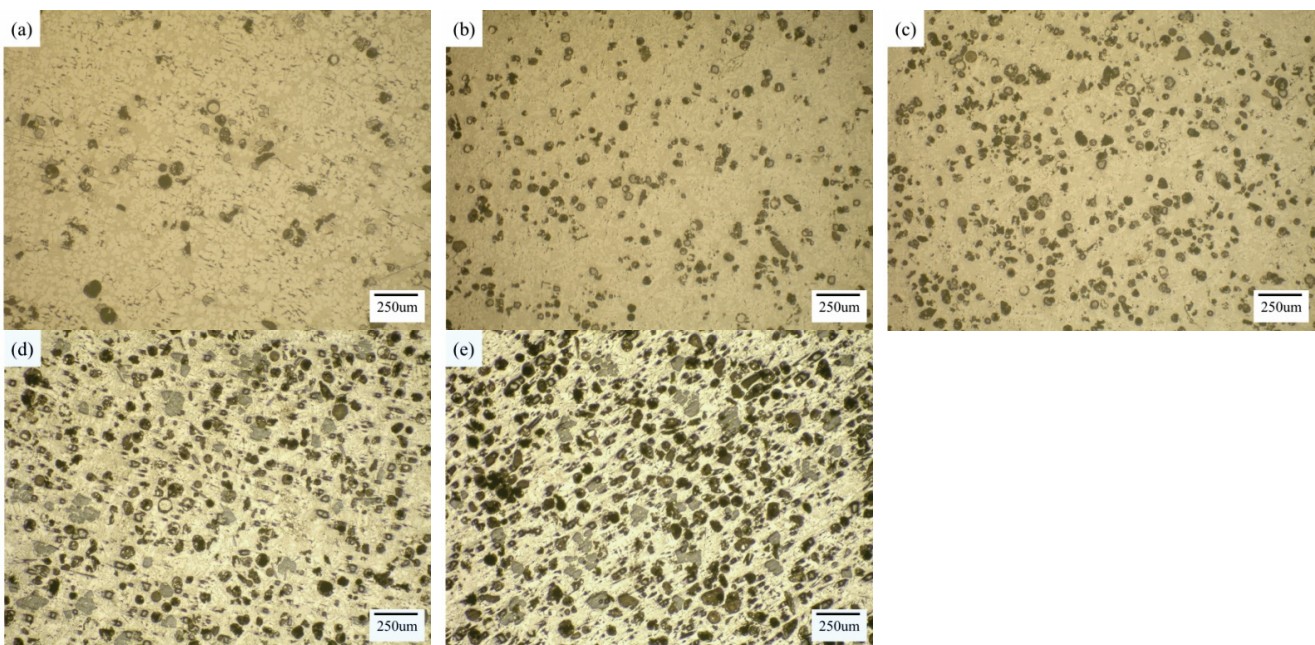

**Figure 5.** Microstructure of aluminum-based fly ash composites (50×): (**a**) 3 *wt*% of fly ash, (**b**) 6 *wt*% of fly ash, (**c**) 9 *wt*% of fly ash, (**d**) 12 *wt*% of fly ash, and (**e**) 15 *wt*% of fly ash.

Figure 6 shows optical micrographs of the ALFA composites prepared with different fly ash addition ratios at 500× magnification. As observed, no obvious gap existed between the fly ash particles (solid phase) and aluminum matrix (liquid phase). The good binding between the solid and liquid phases proves that pretreatment of the fly ash and the addition of pure magnesium to the liquid metal can effectively improve the wettability between the fly ash particles and aluminum matrix, and then obtain good dispersibility [34]. Moreover, after the fly ash was chemically reacted, the particles decomposed into fragments; however, not all the fly ashes reacted, and some of the fly ashes appeared black inside, which was the internal pore of the original fly ash.

Dey et al. reported that $SiO_2$ and $Fe_2O_3$ react with molten aluminum at high temperatures to form $Al_2O_3$ reinforcement phases, as shown in Equations (3) and (4), and the reaction-produced $Al_2O_3$ is used as the reinforced phase of a metal matrix composite to enhance the mechanical properties of the original metal, which are in situ synthesis-reinforced phase composites. During the chemical reaction to produce the $Al_2O_3$-reinforced phase, the fly ash particles gradually break down, allowing the molten aluminum matrix to penetrate the crushing point into the internal pores of the fly ash to fill the defects of the original pores of the fly ash, thereby reducing the porosity of the material [33,35].

$$2Al_{(l)} + \frac{3}{2}SiO_{2(s)} \rightarrow \frac{3}{2}Si_{(s)} + Al_2O_{3(s)} \text{ (931 to 1683 K)}, \tag{3}$$

$$2Al_{(l)} + 2Fe_2O_{3(s)} \rightarrow 2Fe_{(s)} + Al_2O_{3(s)} \text{ (950 to 1033 K)} \tag{4}$$

In the preparation of ALFA composites, the addition of Mg reduces the surface tension of the molten aluminum matrix and enhances its wettability between the matrix alloy and fly ash. Mg also reacts with $Al_2O_3$ and $SiO_2$ in the fly ash to form the magnesium-aluminum spinel ($MgAl_2O_4$) reinforcement phase, as shown in Equations (5) and (6). In addition, Si reacts with Mg by reduction to form a reinforced phase of magnesium silica ($Mg_2Si$), as shown in Equation (7). Although $MgAl_2O_4$ and $Mg_2Si$ can harden the matrix alloy, excess $MgAl_2O_4$ makes the material extremely brittle, whereas $Mg_2Si$ is prone to corrosion [12–15,36,37].

$$3Mg_{(l)} + 4Al_2O_{3(s)} \rightarrow 3MgAl_2O_{4(s)} + 2Al_{(l)} \tag{5}$$

$$2SiO_{2(s)} + Mg_{(l)} + 2Al_{(l)} \rightarrow 2MgAl_2O_{4(s)} + 2Si_{(s)} \tag{6}$$

$$2Mg_{(l)} + Si_{(s)} \rightarrow Mg_2Si_{(s)} \tag{7}$$

When the fly ash reacts with the matrix alloy, the reaction starts from the contact surface and gradually proceeds from the surface to the inside of the fly ash; therefore, the wall thickness of the fly ash becomes thinner and then breaks into small fragments, and the aluminum alloy can penetrate the interior of the fly ash through the gap between the fragments and fill the hollow pores inside it. Matrix alloys cannot penetrate the hollow pores if they are not fully reacted or half-reacted, resulting in defects in the composite. Although hollow pores can reduce the density of the composite, this defect also degrades the mechanical properties and increases the porosity of the composite, resulting in an increase in porosity with an increase in the addition ratio of fly ash content. In the microstructure photos of the composite prepared with each addition ratio of fly ash, it was observed that the structure contained Si flake precipitates reduced by the chemical reaction. When the fly ash was added in an amount of 3 *wt*%, only a few Si flake precipitates were found in the composite. However, when the addition ratio reached 15 *wt*%, the Si flake precipitates of the composite increased significantly, meaning the Si content in the composite increased with the addition ratio of fly ash; when the reduced Si in the matrix reached saturation, Si could not be dissolved and precipitated in the aluminum matrix during the cooling and solidification process. The Si component of the ADC10 aluminum alloy had a fixed weight percentage of approximately 7.5–8.5 *wt*%. The $SiO_2$ in the fly ash reacted with the aluminum matrix to reduce and generate additional Si components. Therefore, the precipitated Si in the composite confirmed that the $SiO_2$ in the fly ash reacted with the aluminum matrix to generate $Al_2O_3$ and Si.

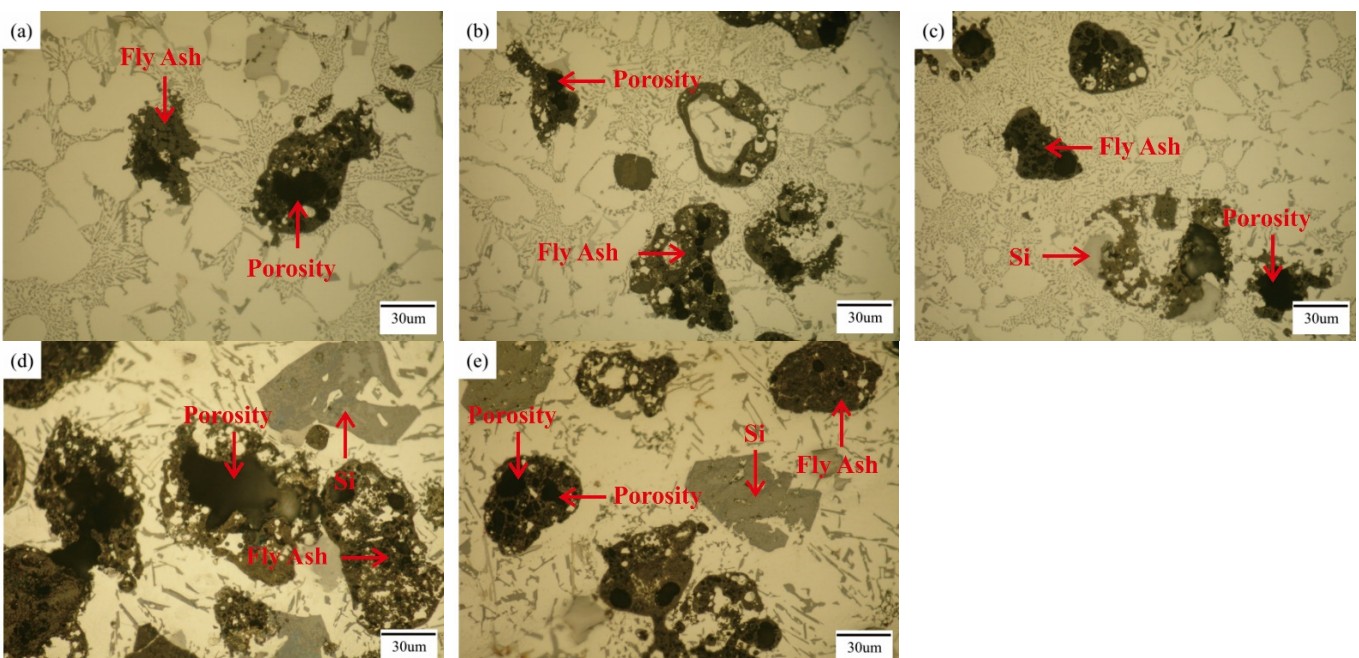

**Figure 6.** Microstructure of aluminum-based fly ash composites (500×): (**a**) 3 *wt*% of fly ash, (**b**) 6 *wt*% of fly ash, (**c**) 9 *wt*% of fly ash, (**d**) 12 *wt*% of fly ash, and (**e**) 15 *wt*% of fly ash.

### 3.2. Density and Porosity

Figures 7 and 8 show the examination results of the density and porosity of the ALFA composite. As shown in Figure 7, the density of the composite decreased as the amount of fly ash increased, indicating that an increased amount of fly ash made the composite lighter. This is because most fly ashes are hollow and porous and their bulk density is lower than that of aluminum matrix. In contrast, the density of the composite decreased

and its porosity increased with the addition ratio of fly ash content, as shown in Figure 8. The ALFA composite was used to perform the chemical reaction at a high temperature of 800 °C for 30 h, although the fly ash was decomposed and broken by the reaction, allowing the aluminum alloy to fill the pores inside the fly ash. However, many pores still existed inside the fly ash that had not been filled by the matrix alloy, which is a reason for the increase in the porosity of aluminum-based fly ash composites. In addition, with a higher addition ratio of fly ash in the ALFA composite preparation process, the fluidity of the slurry reduced and the air involved in the slurry with the stirring operation was more difficult to discharge, resulting in the occurrence of porosity defects in the composite. This was more evident when the addition ratio of fly ash was higher than 9 *wt*%, which caused its porosity to increase rapidly. When the porosity increased, more defects in the composite negatively influenced its mechanical properties.

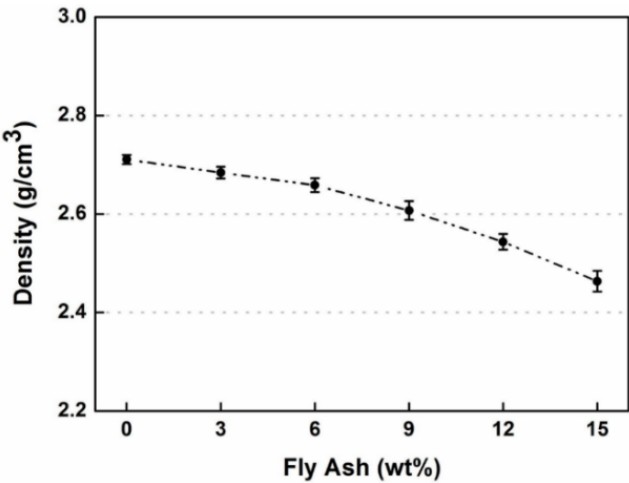

**Figure 7.** Variation in density of different fly ash addition ratios.

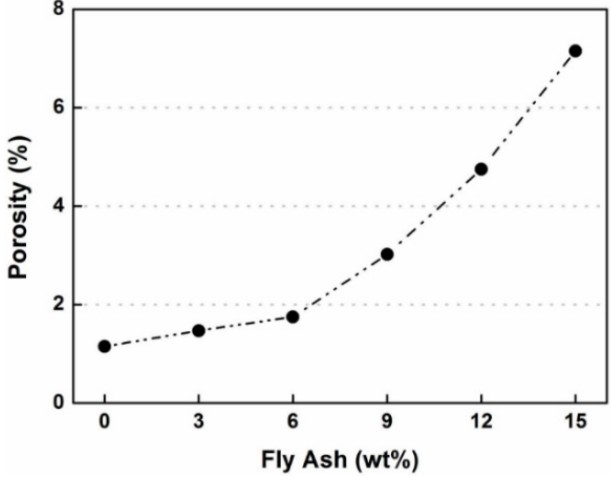

**Figure 8.** Variation in porosity of different fly ash addition ratios.

*3.3. Hardness Test*

The hardness test results of the composite made by each fly ash addition ratio are shown in Figure 9. The hardness was higher than those of the original aluminum matrix when the addition ratios of fly ash were 3, 6, and 9 *wt*%, indicating that the addition of fly ash contributed to an increase in the surface hardness of the composite. However, the hardness decreased when the fly ash addition ratio exceeded 9 *wt*%. According to Myirionis and Kumar [38,39], it can be reasonably inferred that the formation of the $Al_2O_3$ reinforcing phase owing to the reaction between fly ash and aluminum matrix at low fly ash addition

ratios, $Al_2O_3$, and Si, react with Mg to form reinforcing phases such as $MgAl_2O_4$ and $Mg_2Si$, which increase the surface hardness of the composite. Owing to the high quantity of fly ash, the viscosity of the slurry increased when the fly ash addition ratio exceeded 9 *wt*%, and the entrainment of air in the slurry made it difficult to dislodge during the stirring process, resulting in obvious pore defects. Hence, the surface hardness of the composite decreased as the porosity defects increased, and a considerable relationship existed between this trend and porosity. Because the porosity of the composite was less than 2 *wt*%, it had little effect on its hardness; however, when it was over 2 *wt*%, the porosity defects had a dramatic effect on the surface of the composite. At 12 *wt*% and 15 *wt*%, the surface hardness of the composite was lower than that of the matrix alloy. According to the results, the hardness of the ALFA composite with 9 *wt*% of fly ash was highest. It comprises BHN 102, which is 9% higher than that of the original aluminum alloy.

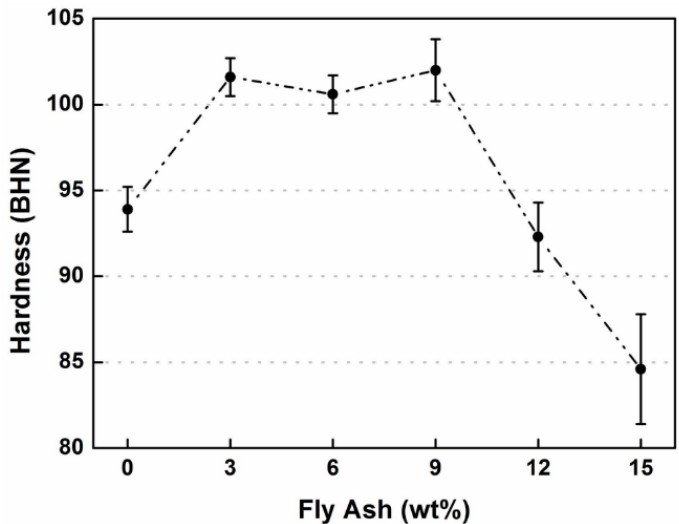

**Figure 9.** Variation in hardness of different fly ash addition ratios.

*3.4. Tensile Test*

The tensile test results for the composites produced with different fly ash addition ratios are shown in Figure 10. In this study, the tensile strength and elongation of the original ADC10 ingots were higher than those of the composites prepared by a chemical reaction at 800 °C for 30 h, especially for the casting prepared with no fly ash addition (0 *wt*%). Compared with the properties of the original ADC10 ingot, the tensile strength and elongation of the ALFA composite showed a significant decrease, which was only half that of ADC10. This result may have been influenced by the small-capacity crucible (2 kg) used in this experiment. During the preparation of the ALFA composite slurry, the degassing operation was not performed and the slag was removed by scraping off the scum on the surface of the slurry that had been reacted for 30 h in the crucible. Consequently, porosities and inclusions may have existed in the ALFA composite, resulting in the failure of the mechanical properties to reach the expected value. Furthermore, compared with castings prepared with no fly ash addition, the tensile strength of composites containing fly ash (3, 6, 9, 12, and 15 *wt*%) was improved because of an increase in the $Al_2O_3$ reinforcing phase generated by the chemical reaction, although $Al_2O_3$ may cause the material to become brittle and less ductile. The best fly ash addition ratio was 6 *wt*% for the ALFA composites. The tensile strength and elongation were 139 MPa and 0.4%, respectively, which were approximately 49% higher and approximately 35% lower than those prepared without fly ash at the same temperature for 30 h, respectively. When the fly ash addition ratio was higher than 9 *wt*%, the tensile strength of the composite decreased significantly owing to the increased porosity of the composite.

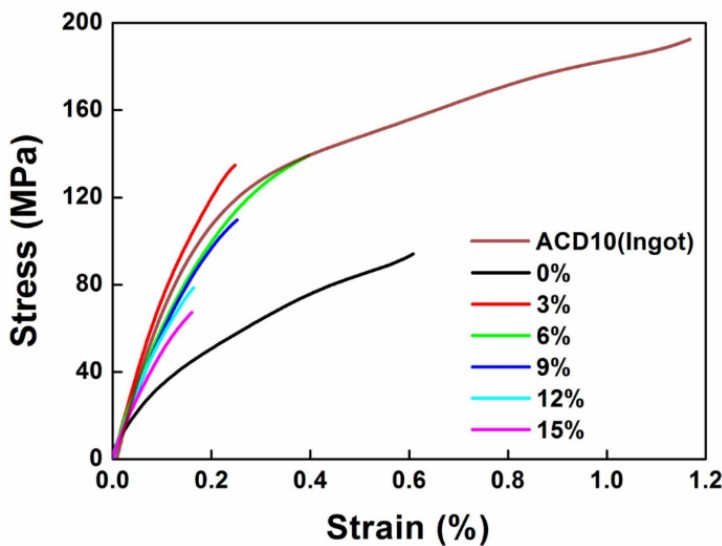

**Figure 10.** Stress–strain diagram for different fly ash addition ratios.

Based on the results of the aforementioned experiments, compared with aluminum matrix, ALFA composites can effectively improve their hardness and tensile strength and also produce lightweight composites; however, when the porosity of the composites is too high, their mechanical properties decrease significantly, even worse than those of the original aluminum matrix. To improve the porosity of composites with high fly ash addition ratios, other processes should be applied to prepare ALFA composites, such as extrusion casting and pressure infiltration casting [10,29], to force the aluminum alloy to penetrate the hollow pores of fly ash and to allow the gas in the slurry to smoothly discharge from the cavities.

### 4. Conclusions

1.  According to the experiments conducted, it was proven that the original silicon content of ADC10 (A380.0) aluminum alloy was approximately 8%; however, the silicon content increased to 10% and more after reacting with fly ash at a temperature of 800 °C for 30 h to form $Al_2O_3$. In the process, Si was reduced and dissolved into the aluminum matrix. The Si content was close to the lower limit of ADC12 (A384.0).
2.  The ALFA composite samples prepared using the stir casting method and the chemical reaction procedure proposed in this study were observed by optical microscopy, which showed that the dispersion of fly ash particles in the ALFA composites was quite good.
3.  The addition of fly ash can effectively reduce the density of ALFA composites. In this study, the Si content in the matrix might have reached the eutectic point (11.7 *wt*%), and the Si could no longer be reacted and reduced when the fly ash addition ratio exceeded 9 *wt*%, increasing the amount of unreacted hollow fly ash. In addition, the air entrained into the slurry during the stirring and pouring process caused the porosity to increase with increasing fly ash content. As the porosity increased, more void defects appeared in the composite, and the adverse effect on its mechanical properties was most significant.
4.  When the fly ash content was higher than 9 *wt*%, the hollow fly ash and the air entrained by stirring could be discharged smoothly. Both hollow fly ash and air defects were observed, which led to a sharp decrease in the hardness and tensile strength of the ALFA composite.
5.  In this study, it was observed that the ALFA composite prepared by adding 6 *wt*% fly ash had the best properties. Its tensile strength, hardness, density, and elongation were 139 MPa, 101 BHN, 2.65 g/cm$^3$, and 0.4%, respectively, which were approximately

49% higher, 7% higher, 2% lower, and 35% lower than those of the casting prepared with no fly ash addition at the same temperature of 800 °C for 30 h, respectively.

6.  Compared to other published articles, it was observed that when the addition of fly ash was less than 9%, the changing trends of density and hardness were consistent; however, when the addition of fly ash was more than 9%, the hardness dropped significantly, whereas the hardness obtained by pressure infiltration casting continued to increase. This is because the viscosity of the slurry increased when the addition amount exceeded 9 *wt*%, and the air entrained in the slurry made it difficult to dislodge during the stirring process, resulting in obvious pore defects and decreasing the surface hardness of the composite material.

7.  The general-purpose die-casting alloy ADC10 was used as the matrix alloy and an amount of Mg was added (contributing to the wettability of the aluminum matrix and fly ash). The Si content of the matrix can be increased using the Si reduced by the reaction of fly ash and aluminum, and the reinforcing phase $Al_2O_3$ is generated simultaneously. The strength and toughness of the strengthened matrix ADC10 can meet the requirements of automotive components without using the more expensive special die-casting alloys.

8.  The specific contribution of this study was not only to reduce industrial waste but also to create the value of waste reuse and improve the mechanical and physical properties of general products.

**Author Contributions:** Conceptualization, S.H.J.; methodology, S.H.J. and C.-F.L.; Experiments, C.-F.L.; Analysis, S.H.J. and C.-F.L.; Verification, S.H.J. All authors have read and agreed to the published version of the manuscript.

**Funding:** Ministry of Science and Technology, Taiwan.

**Institutional Review Board Statement:** Not applicable.

**Informed Consent Statement:** Not applicable.

**Data Availability Statement:** The data to support the findings of this study are included within the article.

**Conflicts of Interest:** The authors declare no conflict of interest.

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
