# Peer review of "Influence of Different Addition Ratios of Fly Ash on Mechanical Properties of ADC10 Aluminum Matrix Composites"

_metals, doi:10.3390/met12040653_

Round 1

Reviewer 1 Report

I suggest for the paper to be published with certain suggestions and proposals

  • It is suggested that the authors cite several papers in the introduction related to the application of metal matrix composites, with a focus on aluminum metal matrix composites. (e.g. 17559/TV-20130905094303, 10.18485/aeletters.2018.3.2.2).
  • It is also suggested that authors cite recent literature because the reference list is very short and the papers are on average about 15 years old.
  • In the introduction, add a review of the research of several new works with a focus on materials whose characteristics are examined in the paper.
  • At the end of the conclusion, emphasize what is the main contribution of the work.
  • Was only one mold or several molds of the same dimensions used for sample preparation?
  • How do the authors ensure that the fly ash is properly distributed in the base material, considering the different mechanical characteristics of the reinforcement and the base material?
  • Was only one sample made for each fly ash content made in one or more? Based on what was that number of samples selected?
  • When the mechanical characteristics of the material were determined, how many samples were analyzed for each fly ash content? Show the diagram of deviations of the obtained values from the mean values.
  • In addition to the shown microstructure, give the EDS analysis of the surface of the samples, which confirms the given data.
  • How do you explain that the hardness of composites first increases and then decreases sharply with increasing mass content of fly ash?
  • Compare the obtained results with other researchers results in this field
  • How different are these results from other results and how do you explain that?

Reviewer 2 Report

1- Although the application of fly ash as an industrial wate materials and agricultural waste materials in metal matrix composite is a old topic but still merits of investigation for the sake of waste recycling and reducing the production cost. The manuscript is well written and well organized, however most of the references used in the introduction section are old and it is recommended that the authors include more recent publications to emphasize the importance of topic. The authors are recommended to use the following publication as well as others: 

  • Development of metal-matrix composites from industrial/agricultural waste materials and their derivatives, Critical Reviews in Environmental Science and Technology, 2016, 46:2, 143-208.
  • "Effect of rice-husk ash on properties of laminated and functionally graded Al/SiC composites by one-step pressureless infiltration", Journal of Alloys and Compounds, 2015, 644, 256-266.
  • "Bilayer graded Al/B4C/rice husk ash composite: Wettability behavior, thermo-mechanical, and electrical properties", Journal of Composite Materials, 2018, 52 (27), 3745-3758.

2- The discussion in the tensile test section has to be more elaborated. 

3- Please provide XRD pattern of some of the composites to confirm the reaction between fly ash and aluminum matrix.

Reviewer 3 Report

The reported results are interesting, but there are a number of observations regarding this manuscript:

1. First, it should be noted that aluminum - fly ash composites have been developed for a very long time (see, for example, the work 10.1007/BF03222635 of 1994, and many others). To date, extensive experience has been accumulated in the development of metallurgical technologies for the production of Al/fly ash composites, the study of their structure and various properties. For reference, see review paper 10.1007/s11837-020-04170-z on this topic. The authors in the Introduction section extremely superficially cover the background of the study. It was necessary to briefly reflect on the achievements reported in numerous relevant works in recent years. At the same time, it is necessary to detail what was specifically achieved in certain works in relation to the goal set in the work, and not to give a simple statement or enumeration of previously completed works. In fact, right now generalization and interpretation of literature data do not correspond to high quality criteria of the journal. The objectives and significance of this study are not clear. In general, it is not obvious what is new in the manuscript in comparison with numerous published works in this area.

2. The statement about the too-high cost of common reinforcing particles (SiC, Al2O3, Gr) as the main factor limiting the industrial use of composites is erroneous. These components are widely available and have long been used by companies producing cast metal matrix composites. Rather, here we should talk about further reducing the cost of cast composites and improving the manufacturability of their production by metallurgical techniques, as well as the effective disposal of industrial waste in the form of fly ash.

3. The description of the mechanism of interaction between the aluminum melt and fly ash should have been moved from the Introduction section to the Discussion section and should be given in a much more extended form. This would make it possible to strengthen the analysis of the obtained results.

4. Methods do not provide enough details for the general reader to repeat the experiments. It is not clear what the actual chemical composition of the matrix alloy was. Table 1 shows not the chemical, but the phase composition of fly ash, and does not give 100% in total (that is, not all components were identified?). How was the phase composition determined? It is not indicated what specific equipment (brand, manufacturer) was used in the research (refers to an optical microscope, hardness tester, tensile testing machine, etc.). Brinell hardness measurement modes are not given (load, indenter diameter). It would be useful to show the morphology of the initial fly ash particles (eg using SEM).

5. Methods and equipment related to metallurgical processes for producing composites are reflected in a very fragmentary way. The type of melting unit, material and capacity of the crucible, material and configuration of the stirrer, material and dimensions of the mold (line 98), etc. are not specified. Was degassing treatment used during melting or any protective atmosphere? There is no consistency in the description of the technique for obtaining composites. It is not clear how the relevant stirring parameters were chosen and worked out, for example, the temperature of the melt at fly ash injection, the speed of the stirrer, the stirring time, etc.

6. Insufficient research tools do not provide a comprehensive picture. Phase constituents on the presented optical micrographs are not analyzed and not discussed. The interaction products of fly ash with aluminum melt should be identified using appropriate tools, at least XRD, and more preferably quantitative EDX analysis. So far, the formation of Al2O3, Mg2Si, MgAl2O4, etc. phases under the conditions of the experiment has not been proven in any way, and no conclusions can be drawn.

7. The key section with the discussion of the results should be radically revised. The manuscript so far contains only a very superficial analysis of the data obtained, and some aspects have not been disclosed at all. There is no description of the mechanisms of strengthening the composite due to fly ash reinforcement. A detailed analysis of the strengthening features should have been carried out (since this was the stated goal of the manuscript), and the relationships between the structure and properties should be explained during this analysis. Here, a study of the fracture surfaces of samples using SEM would be very useful. It is also necessary to compare the data obtained by the authors with previously published studies on this topic.

8. There is no information about the change in the actual chemical and phase composition of the alloy after adding different amounts of fly ash. For example, the reduction of Fe2O3 with liquid aluminum will be accompanied by saturation of the matrix alloy with an iron impurity. It was necessary to show experimentally what limits this saturation can reach under the conditions considered. This, in turn, could also affect the mechanical properties of the composites.

9. The text of the manuscript as a whole contains numerous spelling errors and needs to be completely revised by a proficient English speaker. 

In general, the materials presented are interesting, but not comprehensively processed and explained, that's why cause many questions. The above shortcomings make this paper unacceptable for publication, especially in such a high-level journal. The work as a whole has a low methodological level, therefore it does not make a significant contribution to the relevant field of science and is of no interest to the journal's audience. 

Round 2

Reviewer 2 Report

It is expected that the authors follow the reviewer's recommendation in terms of citing the articles. All three reviewers suggested some articles for including in the manuscript, but the authors refuse mentioning them in the text. Please provide the reason for that.   

Reviewer 3 Report

The manuscript still raises a lot of questions. 
Firstly, it is dissatisfied that the authors did not indicate the corrections made in the manuscript in any way. It is customary to highlight changes in the text of a manuscript (for example, in red) so that the reviewer can clearly see what specific corrections were made by the authors compared to the original version of the manuscript. 
According to most of this reviewer's comments, the authors worked out only superficially or simply ignored them.
More specific: 

1. The Introduction section now looks better. However, the manuscript did not answer what is the essential novelty of the goal and objectives of the research in comparison with the already published works on this topic. See the instructions for the authors of Metals journal: "The introduction should briefly place the study in a broad context and highlight why it is important". 

2. Comment No. 3 of the first round of peer review is not taken into account in any way, and the answer of the authors is unconvincing and not substantiated. The reaction mechanisms should have been moved to the Discussion section and explained in greater detail.

3. The authors did not take into account comment No. 4 properly. For example, table 2 lists the composition of the ADC10 alloy according to the standard, but not the actual composition of the alloy used in this study (as it measured by authors). Brands and manufacturers of equipment used should be included in the text of the manuscript. The same applies to measurement modes. 

4. Reply to comment No. 5 of the first round of peer review cannot be considered satisfactory. The information on technological equipment and modes specified in the answer should be included in the text of the manuscript. There are no answers to the question about finding optimal mixing modes. The configuration (shape) of the mixing device is neither shown nor described, etc.

5. Reply to comment No. 6 is not given and no corrections have been made. There is no XRD or EDX data. Without these data, any conclusions will be invalid.

6. The authors apparently did not fully understand the essence of the comment No. 7. Authors must show how the results can be interpreted from the perspective of previous studies and the working hypotheses. To do this, it is necessary to consider in more detail in Discussion section the mechanisms of interaction between fly ash and aluminum melt (including using the methods according to comment No. 6), and then establish the relationship between the mechanisms of formation of the microstructure and phase composition with the mechanisms of strengthening of the resulting aluminum matrix composites.

Thus, the authors did not pay enough attention to correcting the manuscript according to the comments. However, since the results may still have some practical significance, the review of the manuscript can be continued after careful correction of the previously indicated comments.

Round 3

Reviewer 2 Report

It can be accepted in its current form.

Author Response

Thanks for your affirmation.

Reviewer 3 Report

I believe that the authors have made some efforts to improve the submitted materials and now the manuscript can be recommended for publication after the following corrections:

1. The manuscript contains mistakes in terminology. For example, it is not customary to write the term "aluminum substrate" and even more so "aluminum-based substrate" in relation to aluminum matrix composites, please write "aluminum matrix" or "matrix alloy", etc. Instead of "stirring casting" authors should write "stir casting". "Strengthening phase" is better written as "reinforcing phase". This should be checked throughout the text of the manuscript. 

2. Please decipher the abbreviation L.O.I. (line 111).

3. Mistake on line 179: The Rule of Mixtures allows you to estimate the theoretical density of the composite or some other additive properties, but not the porosity of the composite.

4. The heading of subsection 3.3 (line 291) is obviously redundant ("The study of Mirionis...").

5. I strongly recommend careful proofreading of the manuscript by an English-speaking technical professional.
